# Diffusion Self-Guidance for Controllable Image Generation

**Dave Epstein**[1,2]   **Allan Jabri**[1]   **Ben Poole**[2]   **Alexei A. Efros**[1]   **Aleksander Holynski**[1,2]
[1]UC Berkeley    [2]Google Research

## Abstract

Large-scale generative models are capable of producing high-quality images from detailed text descriptions. However, many aspects of an image are difficult or impossible to convey through text. We introduce self-guidance, a method that provides greater control over generated images by guiding the internal representations of diffusion models. We demonstrate that properties such as the shape, location, and appearance of objects can be extracted from these representations and used to steer the sampling process. Self-guidance operates similarly to standard classifier guidance, but uses signals present in the pretrained model itself, requiring no additional models or training. We show how a simple set of properties can be composed to perform challenging image manipulations, such as modifying the position or size of specific objects, merging the appearance of objects in one image with the layout of another, composing objects from multiple images into one, and more. We also show that self-guidance can be used for editing real images. See our project page for results and an interactive demo: https://dave.ml/selfguidance

## 1   Introduction

Generative image models have improved rapidly in recent years with the adoption of large text-image datasets and scalable architectures [7, 10, 12, 27, 28, 32, 34, 38]. These models are able to create realistic images given a text prompt describing just about anything. However, despite the incredible abilities of these systems, discovering the right prompt to generate the *exact* image a user has in mind can be surprisingly challenging. A key issue is that all desired aspects of an image must be communicated through text, even those that are difficult or even impossible to convey precisely.

To address this limitation, previous work has introduced methods [9, 14, 18, 30] that tune pretrained models to better control details that a user has in mind. These details are often supplied in the form of reference images along with a new textual prompt [2, 4] or other forms of conditioning [1, 31, 39]. However, these approaches all either rely on fine-tuning with expensive paired data (thus limiting the scope of possible edits) or must undergo a costly optimization process to perform the few manipulations they are designed for. While some methods [11, 21, 22, 36] can perform zero-shot editing of an input image using a target caption describing the output, these methods only allow for limited control, often restricted to structure-preserving appearance manipulation or uncontrolled image-to-image translation.

By consequence, many simple edits still remain out of reach. For example, how can we move or resize one object in a scene without changing anything else? How can we take the appearance of an object in one image and copy it over to another, or combine the layout of one scene with the appearance of a second one? How can we generate images with certain items having precise shapes at specific positions on the canvas? This degree of control has been explored in the past in smaller scale settings [5, 8, 19, 24, 37, 40], but has not been convincingly demonstrated with modern large-scale diffusion models [26, 32, 38].

37th Conference on Neural Information Processing Systems (NeurIPS 2023).

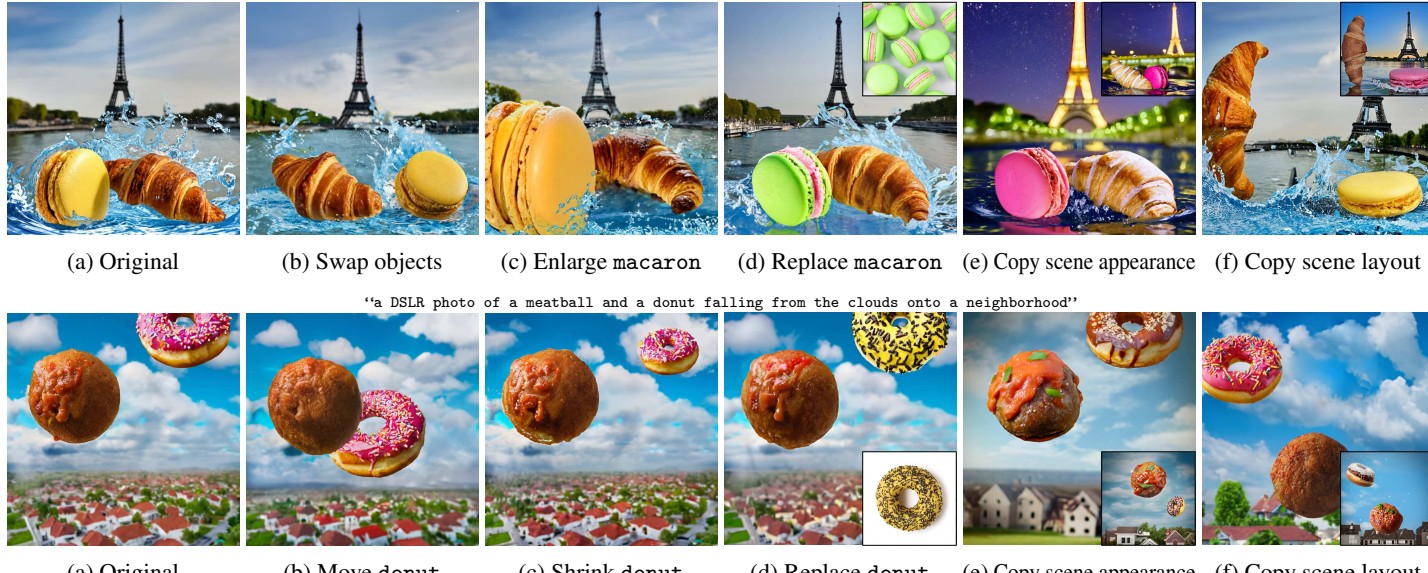

"a photo of a giant macaron and a croissant splashing in the Seine with the Eiffel Tower in the background"

| (a) Original | (b) Swap objects | (c) Enlarge `macaron` | (d) Replace `macaron` | (e) Copy scene appearance | (f) Copy scene layout |

"a DSLR photo of a meatball and a donut falling from the clouds onto a neighborhood"

| (a) Original | (b) Move `donut` | (c) Shrink `donut` | (d) Replace `donut` | (e) Copy scene appearance | (f) Copy scene layout |

Figure 1: **Self-guidance** is a method for controllable image generation that guides sampling using the attention and activations of a pretrained diffusion model. With self-guidance, we can move or resize objects, or even replace them with items from real images, without changing the rest of the scene (b-d). We can also borrow the appearance of other images or rearrange scenes into new layouts (e-f).

We propose *self-guidance*, a zero-shot approach which allows for direct control of the shape, position, and appearance of objects in generated images. Self-guidance leverages the rich representations learned by pretrained text-to-image diffusion models – namely, intermediate activations and attention – to steer attributes of entities and interactions between them. These constraints can be user-specified or transferred from other images, and rely only on knowledge internal to the diffusion model. Through a variety of challenging image manipulations, we demonstrate that self-guidance using only a few simple properties allows for granular, disentangled manipulation of the contents of generated images (Figure 1). Further, we show that self-guidance can also be used to reconstruct and edit real images.

Our key contributions are as follows:

- We introduce *self-guidance*, which takes advantage of the internal representations of pretrained text-to-image diffusion models to provide disentangled, zero-shot control over the generative process without requiring auxiliary models or supervision.

- We find that properties such as the size, location, shape, and appearance of objects can be extracted from these representations and used to meaningfully guide sampling in a zero-shot manner.

- We demonstrate that this small set of properties, when composed, allows for a wide variety of surprisingly complex image manipulations, including control of relationships between objects and the way modifiers bind to them.

- Finally, by reconstructing captioned images using their layout and appearance as computed by self-guidance, we show that we can extend our method to editing real images.

## 2 Background

### 2.1 Diffusion generative models

Diffusion models learn to transform random noise into high-resolution images through a sequential sampling process [12, 33, 35]. This sampling process aims to reverse a fixed time-dependent destructive process that corrupts data by adding noise. The learned component of a diffusion model is a neural network $\epsilon_\theta$ that tries to estimate the denoised image, or equivalently the noise $\epsilon_t$ that was added to create the noisy image $z_t = \alpha_t x + \sigma_t \epsilon_t$. This network is trained with loss:

$$L(\theta) = \mathbb{E}_{t \sim \mathcal{U}(1,T), \epsilon_t \sim \mathcal{N}(0,\mathbf{I})} \left[ w(t) ||\epsilon_t - \epsilon_\theta(z_t; t, y)||^2 \right], \tag{1}$$

where $y$ is an additional conditioning signal like text, and $w(t)$ is a function weighing the contributions of denoising tasks to the training objective, commonly set to 1 [12, 15]. A common choice for $\epsilon_\theta$ is a U-Net architecture with self- and cross-attention at multiple resolutions to attend to conditioning text in $y$ [28, 29, 32]. Diffusion models are score-based models, where $\epsilon_\theta$ can be seen as an estimate of the score function for the noisy marginal distributions: $\epsilon_\theta(z_t) \approx -\sigma_t \nabla_{z_t} \log p(z_t)$ [35].

Given a trained model, we can generate samples given conditioning $y$ by starting from noise $z_T \sim \mathcal{N}(0, I)$, and then alternating between estimating the noise component and updating the noisy image:

$$\hat{\epsilon}_t = \epsilon_\theta(z_t; t, y), \quad z_{t-1} = \text{update}(z_t, \hat{\epsilon}_t, t, t-1, \epsilon_{t-1}), \tag{2}$$

where the update could be based on DDPM [12], DDIM [34], or another sampling method (see Appendix for details). Unfortunately, naïvely sampling from conditional diffusion models does not produce high-quality images that correspond well to the conditioning $y$. Instead, additional techniques are utilized to modify the sampling process by altering the update direction $\hat{\epsilon}_t$.

## 2.2 Guidance

A key capability of diffusion models is the ability to adapt outputs after training by *guiding* the sampling process. From the score-based perspective, we can think of guidance as composing score functions to sample from richer distributions or to introduce conditioning on auxiliary information [7, 17, 35]. In practice, using guidance involves altering the update direction $\hat{\epsilon}_t$ at each iteration.

*Classifier guidance* can generate conditional samples from an unconditional model by combining the unconditional score function for $p(z_t)$ with a classifier $p(y|z_t)$ to generate samples from $p(z_t|y) \propto p(y|z_t)p(z_t)$ [7, 35]. To use classifier guidance, one needs access to a labeled dataset and has to learn a noise-dependent classifier $p(y|z_t)$ that can be differentiated with respect to the noisy image $z_t$. While sampling, we can incorporate classifier guidance by modifying $\hat{\epsilon}_t$:

$$\hat{\epsilon}_t = \epsilon_\theta(z_t; t, y) - s\sigma_t \nabla_{z_t} \log p(y|z_t), \tag{3}$$

where $s$ is an additional parameter controlling the guidance strength. Classifier guidance moves the sampling process towards images that are more likely according to the classifier [7], achieving a similar effect to truncation in GANs [3], and can also be applied with pretrained classifiers by first denoising the intermediate noisy image (though this requires additional approximations [1]).

In general, we can use any energy function $g(z_t; t, y)$ to guide the diffusion sampling process, not just the probabilities from a classifier. $g$ could be the approximate energy from another model [17], a similarity score from a CLIP model [23], an arbitrary time-independent energy as in universal guidance [1], bounding box penalties on attention [6], or any attributes of the noisy images. We can incorporate this additional guidance alongside *classifier-free guidance* [13] to obtain high-quality text-to-image samples that also have low energy according to $g$:

$$\hat{\epsilon}_t = (1 + s)\epsilon_\theta(z_t; t, y) - s\epsilon_\theta(z_t; t, \emptyset) + v\sigma_t \nabla_{z_t} g(z_t; t, y), \tag{4}$$

where $s$ is the classifier-free guidance strength and $v$ is an additional guidance weight for $g$. As with classifier guidance, we scale by $\sigma_t$ to convert the score function to a prediction of $\epsilon_t$. The main contribution of our work is to identify energy functions $g$ useful for controlling properties of objects and interactions between them.

## 2.3 Where can we find signal for controlling diffusion?

While guidance is a flexible way of controlling the sampling process, energy functions typically used [1, 39] require auxiliary models (adapted to be noise-dependent) as well as data annotated with properties we would like to control. Can we circumvent these costs? Recent work [11, 36] has shown that the intermediate outputs of the diffusion U-Net encode valuable information [16, 25] about the structure and content of the generated images. In particular, the self and cross-attention maps $\{\mathcal{A}_{i,t} \in \mathbb{R}^{H_i \times W_i \times K}\}$ often encode structural information [11] about object position and shape, while the network activations $\{\Psi_{i,t} \in \mathbb{R}^{H_i \times W_i \times D_i}\}$ allow for maintaining coarse appearance [36] when extracted from appropriate layers. While these editing approaches typically share attention and activations naively between subsequent sampling passes, drastically limiting the scope of possible manipulations, we ask: what if we tried to harness model internals in a more nuanced way?

Figure 2: **Overview:** We leverage representations learned by text-image diffusion models to steer generation with *self-guidance*. By constraining intermediate activations $\Psi_t$ and attention interactions $\mathcal{A}_t$, self-guidance can control properties of entities named in the prompt. For example, we can change the position and shape of the burger, or copy the appearance of ice cream from a source image.

# 3 Self-guidance

Inspired by the rich representations learned by diffusion models, we propose self-guidance, which places constraints on intermediate activations and attention maps to steer the sampling process and control entities named in text prompts (see Fig. 2). These constraints can be user-specified or copied from existing images, and rely only on knowledge internal to the diffusion model.

We identify a number of properties useful for meaningfully controlling generated images, derived from the set of softmax-normalized attention matrices $\left\{\mathcal{A}_{i,t} \in \mathbb{R}^{H_i \times W_i \times K}\right\}$ and activations $\left\{\Psi_{i,t} \in \mathbb{R}^{H_i \times W_i \times D_i}\right\}$ extracted from the standard denoising forward pass $\epsilon_\theta(z_t; t, y)$. To control an object mentioned in the text conditioning $y$ at token indices $k$, we can manipulate the corresponding attention channel(s) $\mathcal{A}_{i,t,\cdot,\cdot,k} \in \mathbb{R}^{H_i \times W_i \times |k|}$ and activations $\Psi_{i,t}$ (extracted at timestep $t$ from a noisy image $z_t$ given text conditioning $y$) by adding guidance terms to Eqn. 4.

**Object position.** To represent the position of an object (omitting attention layer index and timestep for conciseness), we find the center of mass of each relevant attention channel:

$$\texttt{centroid}\,(k) = \frac{1}{\sum_{h,w} \mathcal{A}_{h,w,k}} \left[ \frac{\sum_{h,w} w \cdot \mathcal{A}_{h,w,k}}{\sum_{h,w} h \cdot \mathcal{A}_{h,w,k}} \right] \tag{5}$$

We can use this property to guide an object to an absolute target position on the image. For example, to move "$\texttt{burger}$" to position $(0.3, 0.5)$, we can minimize $\|(0.3, 0.5) - \texttt{centroid}\,(k)\|_1$. We can also perform a relative transformation, e.g., move "$\texttt{burger}$" to the right by $(0.1, 0.0)$ by minimizing $\|\texttt{centroid}_{\text{orig}}\,(k) + (0.1, 0.0) - \texttt{centroid}\,(k)\|_1$.

**Object size.** To compute an object's size, we spatially sum its corresponding attention channel:

$$\texttt{size}\,(k) = \frac{1}{HW} \sum_{h,w} \mathcal{A}_{h,w,k} \tag{6}$$

In practice, we find it beneficial to differentiably threshold the attention map $\mathcal{A}_{\text{thresh}}$ before computing its size, to eliminate the effect of background noise. We do this by taking a soft threshold at the midpoint of the per-channel minimum and maximum values (see Appendix for details). As with position, one can guide to an absolute size (*e.g.* half the canvas) or a relative one (*e.g.* 10% larger).

**Object shape.** For even more granular control than position and size, we can represent the object's exact shape directly through the thresholded attention map itself:

$$\texttt{shape}(k) = \mathcal{A}_k^{\text{thresh}} \tag{7}$$

This shape can then be guided to match a specified binary mask (either provided by a user or extracted from the attention from another image) with $\|\texttt{target\_shape} - \texttt{shape}\,(k)\|_1$. Note that we can

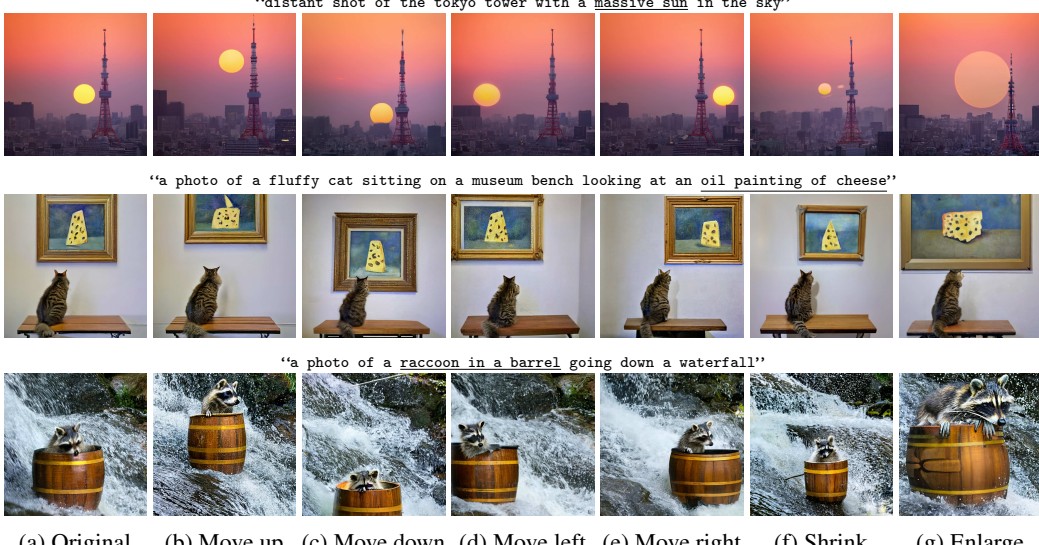

(a) Original   (b) Move up  (c) Move down  (d) Move left  (e) Move right   (f) Shrink   (g) Enlarge

Figure 3: **Moving and resizing objects.** By only changing the properties of one object (as in Eqn. 9), we can move or resize that object without modifying the rest of the image. In these examples, we modify "`massive sun`", "`oil painting of cheese`", and "`raccoon in a barrel`", respectively.

apply any arbitrary transformation (scale, rotation, translation) to this shape before using it as a guidance target, which allows us to manipulate objects while maintaining their silhouette.

**Object appearance.** Considering thresholded attention a rough proxy for object extent, and spatial activation maps as representing local appearance (since they ultimately must be decoded into an unnoised RGB image), we can reach a notion of object-level appearance by combining the two:

$$\texttt{appearance}(k) = \frac{\sum_{h,w}\texttt{shape}(k)\odot\Psi}{\sum_{h,w}\texttt{shape}(k)} \tag{8}$$

## 4   Composing self-guidance properties

The small set of properties introduced in Section 3 can be composed to perform a wide range of image manipulations, including many that are intractable through text. We showcase this collection of manipulations and, when possible, compare to prior work that accomplishes similar effects. All experiments were performed on Imagen [32], producing $1024 \times 1024$ samples. For more samples and details on the implementation of self-guidance, please see the Appendix.

**Adjusting specific properties.** By guiding one property to change and all others to keep their original values, we can modify single objects in isolation (Fig. 3b-3e). For a caption $C = y$ with words at indices $\{c_i\}$, in which $O = \{o_j\} \subseteq C$ are objects, we can move an object $o_k$ at time $t$ with:

$$
\begin{aligned}
g = w_0 &\overbrace{\frac{1}{|O|-1}\sum_{o\neq o_k\in O}\frac{1}{|\mathcal{A}|}\sum_{i=0}^{|\mathcal{A}|}\|\texttt{shape}_{i,t,\text{orig}}(o) - \texttt{shape}_{i,t}(o)\|_1}^{\text{Fix all other object shapes}} \\
+ w_1 &\overbrace{\frac{1}{|O|}\sum_{o\in O}\|\texttt{appearance}_{t,\text{orig}}(o) - \texttt{appearance}_t(o)\|_1}^{\text{Fix all appearances}} \\
+ w_2 &\overbrace{\frac{1}{|\mathcal{A}|}\sum_{i=0}^{|\mathcal{A}|}\left\|\mathcal{T}\left(\texttt{shape}_{i,t,\text{orig}}(o_k)\right) - \texttt{shape}_{i,t}(o_k)\right\|_1}^{\text{Guide } o_k\text{'s shape to translated original shape}}
\end{aligned} \tag{9}
$$

Where $\texttt{shape}_{\text{orig}}$ and $\texttt{shape}$ are extracted from the generation of the initial and edited image, respectively. Critically, $\mathcal{T}$ lets us define whatever transformation of the $H_i \times W_i$ spatial attention map we

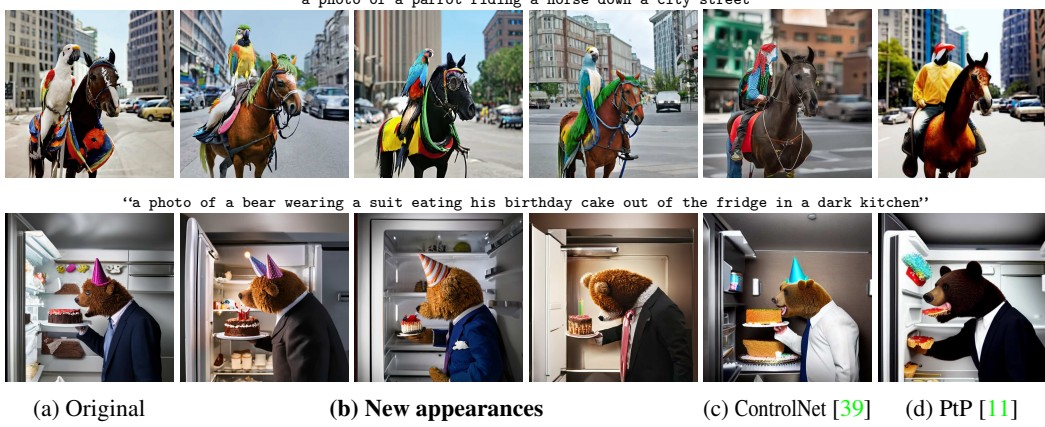

"a photo of a parrot riding a horse down a city street"

"a photo of a bear wearing a suit eating his birthday cake out of the fridge in a dark kitchen"

(a) Original **(b) New appearances** (c) ControlNet [39] (d) PtP [11]

Figure 4: **Sampling new appearances.** By guiding object shapes (Eqn. 7) towards reconstruction of a given image's layout (a), we can sample new appearances for a given scene (b-d).

want. To move an object, $\mathcal{T}$ translates the attention mask the desired amount. We can also resize objects (Fig. 3f-3g) with Eqn. 9 by changing $\mathcal{T}$ to up- or down-sample shape matrices.

Constraining per-object layout but not appearance finds new "styles" for the same scene (Fig. 4):

$$g = w_0 \overbrace{\frac{1}{|O|} \sum_{o \in O} \frac{1}{|\mathcal{A}|} \sum_{i=0}^{|\mathcal{A}|} \|\texttt{shape}_{i,t,\text{orig}}(o) - \texttt{shape}_{i,t}(o)\|_1}^{\text{Fix all object shapes}} \tag{10}$$

We can alternatively choose to guide all words, not just nouns or objects, changing summands to $\sum_{c \neq o_k \in C}$ instead of $\sum_{c \neq o_k \in O}$. See Appendix for further discussion.

**Composition between images.** We can compose properties across multiple images into a cohesive sample, *e.g.* the layout of an image $A$ with the appearance of objects in another image $B$ (Fig. 5):

$$g = w_0 \overbrace{\frac{1}{|O|} \sum_{o \in O} \frac{1}{|\mathcal{A}|} \sum_{i=0}^{|\mathcal{A}|} \|\texttt{shape}_{i,t,A}(o) - \texttt{shape}_{i,t}(o)\|_1}^{\text{Copy object shapes from A}}$$

$$+ w_1 \underbrace{\frac{1}{|O|} \sum_{o \in O} \|\texttt{appearance}_{t,B}(o) - \texttt{appearance}_t(o)\|_1}_{\text{Copy object appearance from B}} \tag{11}$$

We can also borrow only appearances, dropping the first term to sample new arrangements for the same objects, as in the last two columns of Figure 5.

Highlighting the compositionality of self-guidance terms, we can further inherit the appearance and/or shape of objects from several images and combine them into one (Fig. 6). Say we have $J$ images, where we are interested in keeping a single object $o_{k_j}$ from each one. We can collage these objects "in-place" – *i.e.* maintaining their shape, size, position, and appearance – straightforwardly:

$$g = w_0 \overbrace{\frac{1}{J} \sum_j \frac{1}{|\mathcal{A}|} \sum_{i=0}^{|\mathcal{A}|} \|\texttt{shape}_{i,t,j}(o_{k_j}) - \texttt{shape}_{i,t}(o_k)\|_1}^{\text{Copy each object's shape, position, and size}}$$

$$+ w_1 \underbrace{\frac{1}{J} \sum_j \|\texttt{appearance}_{t,j}(o_{k_j}) - \texttt{appearance}_t(o_k)\|_1}_{\text{Copy each object's appearance}} \tag{12}$$

We can also take only the appearances of the objects from these images and copy the layout from another image, useful if object positions in the $J$ images are not mutually compatible (Fig. 6f).

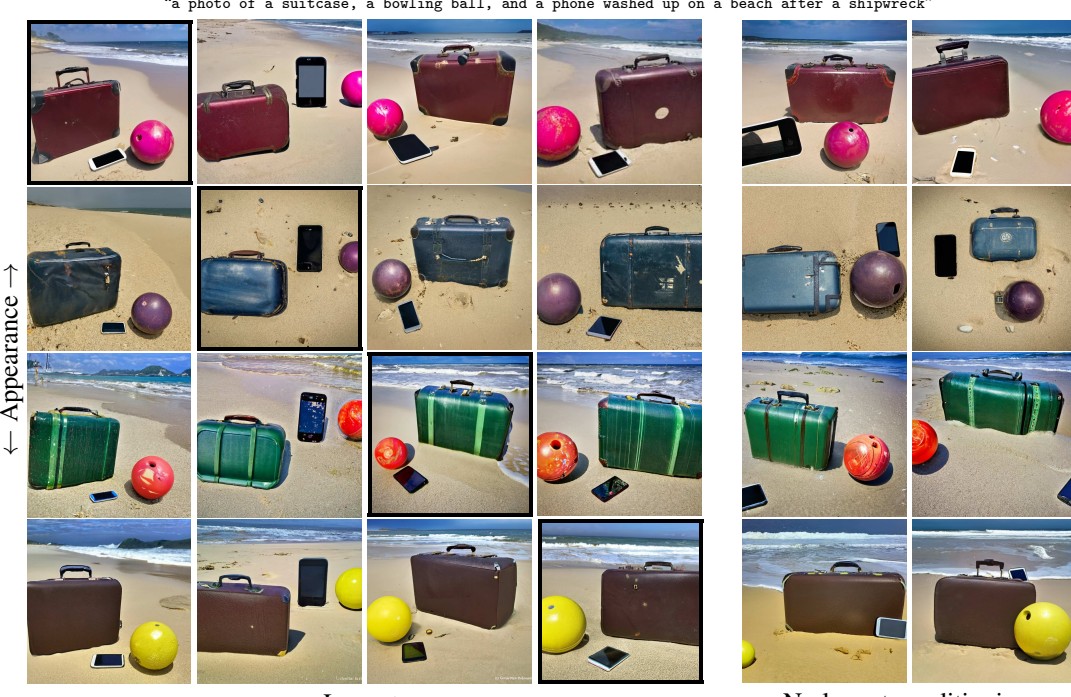

Figure 5: **Mix-and-match.** By guiding samples to take object shapes from one image and appearance from another (Eqn. 11), we can rearrange images into layouts from other scenes. Input images are along the diagonal. We can also sample new layouts of a scene by only guiding appearance (right).

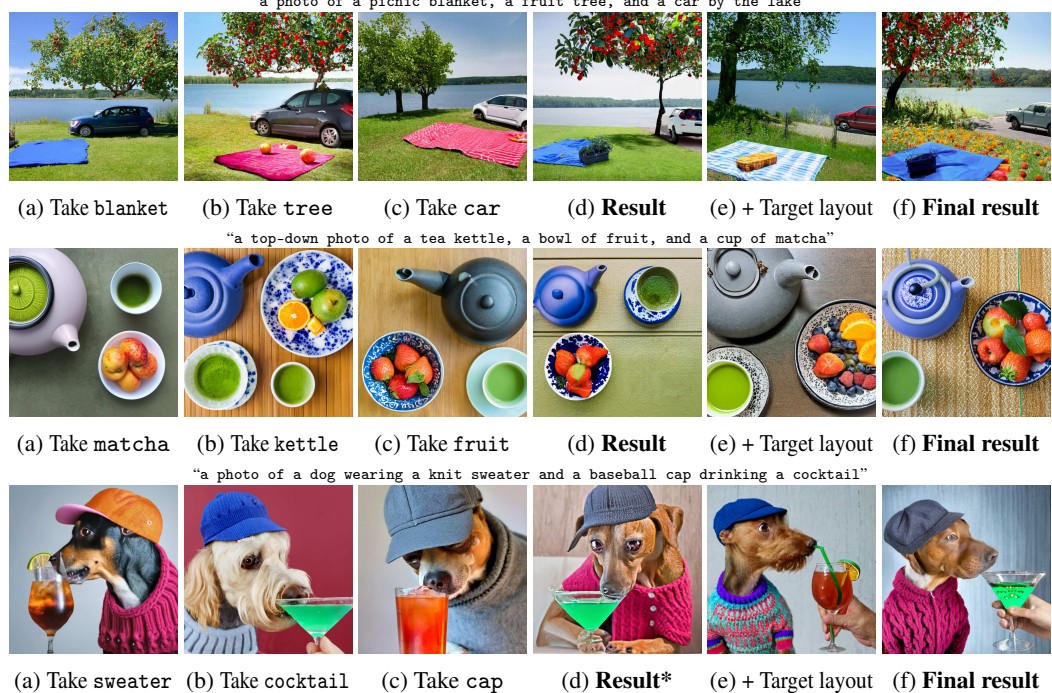

Figure 6: **Compositional generation.** A new scene (d) can be created by collaging objects from multiple images (Eqn. 12). Alternatively – *e.g.* if objects cannot be combined at their original locations due to incompatibilities in these images' layouts (*as in the bottom row) – we can borrow only their appearance, and specify layout with a new image (e) to produce a composition (f) (Eqn. 19).

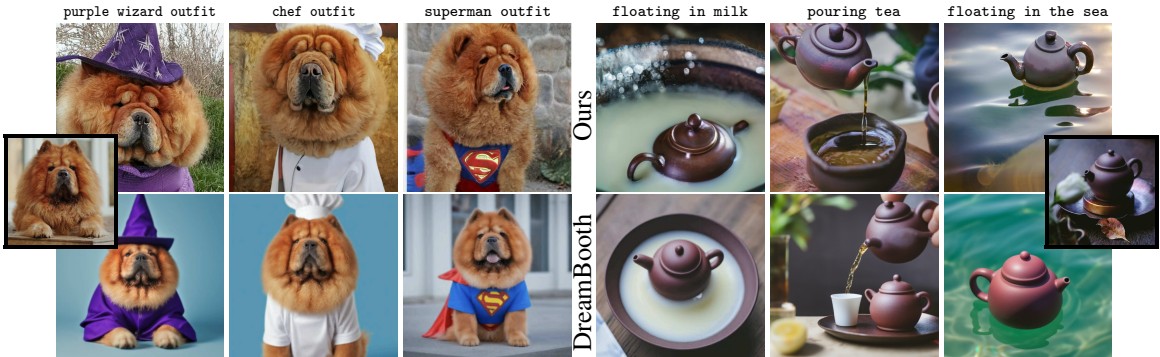

Figure 7: **Appearance transfer from real images.** By guiding the appearance of a generated object to match that of one in a real image (outlined) as in Eqn. 13, we can create scenes depicting an object from real life, similar to DreamBooth [30], but *without any fine-tuning and only using one image*.

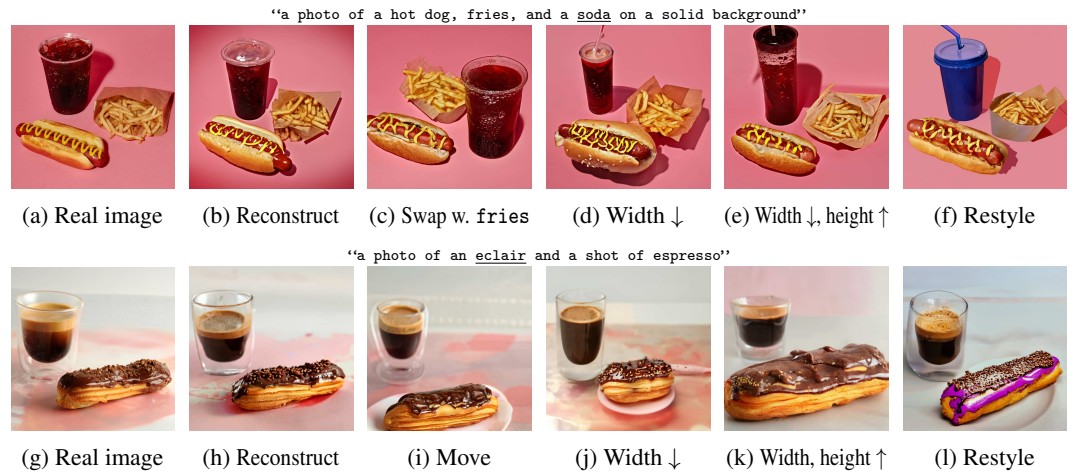

"a photo of a hot dog, fries, and a soda on a solid background"

(a) Real image    (b) Reconstruct    (c) Swap w. `fries`    (d) Width ↓    (e) Width ↓, height ↑    (f) Restyle

"a photo of an eclair and a shot of espresso"

(g) Real image    (h) Reconstruct    (i) Move    (j) Width ↓    (k) Width, height ↑    (l) Restyle

Figure 8: **Real image editing.** Our method enables the spatial manipulation of objects (shown in Figure 3 for generated images) for *real* images as well.

**Editing with real images.** Our approach is not limited to only images generated by a model, whose internals we have access to by definition. By running $T$ noised versions of a (captioned) existing image through a denoiser – one for each forward process timestep – we extract a set of intermediates that can be treated as if it came from a reverse sampling process (see Appendix for more details). In Fig. 8, we show that, by guiding shape and appearance for all tokens, we generate faithful reconstructions of real images. More importantly, we can manipulate these real images just as we can generated ones, successfully controlling properties such as appearance, position, or size. We can also transfer the appearance of an object of interest into new contexts (Fig. 7), from only one source image, and without any fine-tuning:

$$g = w_0 \overbrace{\left\| \texttt{appearance}_{t,\text{orig}}(o_{k_{\text{orig}}}) - \texttt{appearance}_t(o_k) \right\|_1}^{\text{Copy object appearance}} \tag{13}$$

**Attributes and interactions.** So far we have focused only on the manipulation of objects, but we can apply our method to any concept in the image, as long as it appears in the caption. We demonstrate manipulation of verbs and adjectives in Fig. 9, and show an example where certain self-guidance constraints can help in enforcing attribute binding in the generation process.

## 5 Discussion

We introduce a method for guiding the diffusion sampling process to satisfy properties derived from the attention maps and activations within the denoising model itself. While we propose a number of

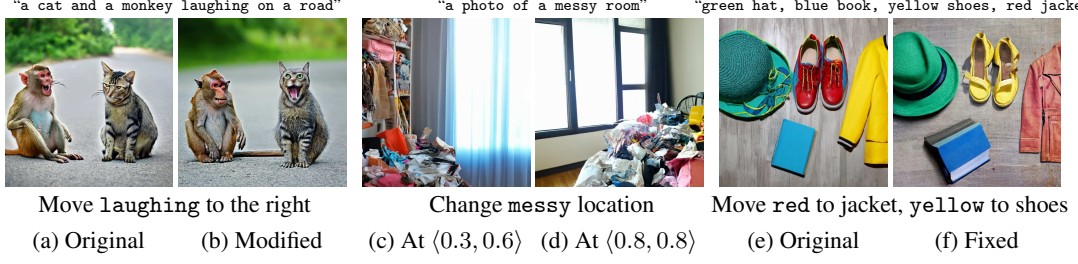

"a cat and a monkey laughing on a road"  "a photo of a messy room"  "green hat, blue book, yellow shoes, red jacket"

Move `laughing` to the right   Change `messy` location   Move `red` to jacket, `yellow` to shoes

(a) Original   (b) Modified   (c) At $\langle 0.3, 0.6 \rangle$   (d) At $\langle 0.8, 0.8 \rangle$   (e) Original   (f) Fixed

Figure 9: **Manipulating non-objects.** The properties of any word in the input prompt can be manipulated, not only nouns. Here, we show examples of relocating adjectives and verbs. The last example shows a case in which additional self-guidance can correct improper attribute binding.

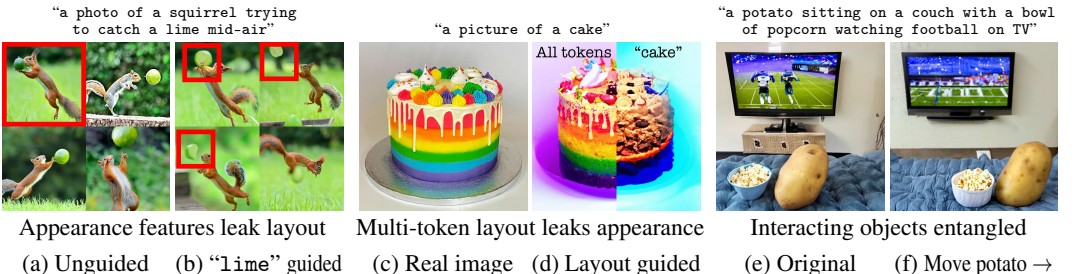

"a photo of a squirrel trying to catch a lime mid-air"   "a picture of a cake"   "a potato sitting on a couch with a bowl of popcorn watching football on TV"

Appearance features leak layout   Multi-token layout leaks appearance   Interacting objects entangled

(a) Unguided   (b) "`lime`" guided   (c) Real image   (d) Layout guided   (e) Original   (f) Move potato →

Figure 10: **Limitations.** Setting high guidance weights for appearance terms tends to introduce unwanted leakage of object position (a-b). Similarly, while heavily guiding the shape of one word simply matches that object's layout as expected, high guidance on the shapes of all tokens results in a leak of appearance information (c-d). Finally, in some cases, objects are entangled in attention space, making it difficult to control them independently (e-f).

such properties, many more certainly exist, as do alternative formulations of those presented in this paper. Among the proposed collection of properties, a few limitations stand out.

The reliance on cross-attention maps imposes restrictions by construction, precluding control over any object that is not described in the conditioning text prompt and hindering fully disentangled control between interacting objects due to correlations in attention maps (Fig. 10e-10f). Selectively applying attention guidance at certain layers or timesteps may result in more effective disentanglement.

Our experiments also show that appearance features often contain undesirable information about spatial layout (Fig. 10a-10b), perhaps since the model has access to positional information in its architecture. The reverse is also sometimes true: guiding the shape of multiple tokens occasionally betrays the appearance of an object (Fig. 10c-10d), implying that hidden high-frequency patterns arising from interaction between attention channels may be used to encode appearance. These findings suggest that our method could serve as a window into the inner workings of diffusion models and provide valuable experimental evidence to inform future research.

## Broader impact

The use-cases showcased in this paper, while transformative for creative uses, carry the risk of producing harmful content that can negatively impact society. In particular, self-guidance allows for a level of control over the generation process that might enable potentially harmful image manipulations, such as pulling in the appearance or layout from real images into arbitrary generated content (e.g., as in Fig. 7). One such dangerous manipulation might be the injection of a public figure into an image containing illicit activity. In our experiments, we mitigate this risk by deliberately refraining from generating images containing humans. Additional safeguards against these risks include methods for embedded watermarking [20] and automated systems for safe filtering of generated imagery.

## Acknowledgements

We thank Oliver Wang, Jason Baldridge, Lucy Chai, and Minyoung Huh for their helpful comments. Dave is supported by the PD Soros Fellowship. Dave and Allan conducted part of this research at Google, with additional funding provided by DARPA MCS and ONR MURI.

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

# Appendix

## A. Implementation details

We apply our self-guidance term following best practices for classifier-free guidance on Imagen [32]. Specifically, where $N$ is the number of DDPM steps, we take the first $\frac{3N}{16}$ steps with self-guidance and the last $\frac{N}{32}$ without. The remaining $\frac{25N}{32}$ steps are alternated between using self-guidance and not using it. We use $N = 1024$ steps. Our method works with 256 and 512 steps as well, though self-guidance weights occasionally require adjustment. We set $v = 7500$ in Eqn. 4 as an overall scale for gradients of the functions $g$ defined below — we find that the magnitude of per-pixel gradients is quite small (often in the range of $10^{-7}$ to $10^{-6}$, so such a high weight is needed to induce changes).

We apply `centroid`, `size`, and `shape` terms on *all cross-attention interactions* in the model we use. In total, there are 36 of these, across the encoder, bottleneck, and decoder, at $8 \times 8$, $16 \times 16$, and $32 \times 32$ resolutions. We apply the `appearance` term using the activations of the penultimate layer in the decoder (two layers before the prediction readout) and the final cross-attention operation. We experimented with features from other parts of the U-Net denoiser, namely early in the encoder before positional information can propagate through the image (to prevent appearance-layout entanglement), but found these to work significantly worse. To avoid degenerate solutions, we apply a stop-gradient to the attention in the `appearance` term so only information about activations is back-propagated. We take the mean spatially of all `shape` terms and across activation dimensions for all `appearance` terms, which we omit in all equations for conciseness.

**Attention mask binarization.** In practice, it is beneficial to differentiably binarize the attention map (with sharpness controlled by $s$) before computing its size or utilizing its shape, to eliminate the effect of background noise (this is empirically less important when guiding centroids, so we do not binarize in that case). We do this by taking a soft threshold at the midpoint of the per-channel minimum and maximum values. More specifically, we apply a shifted sigmoid on the attention normalized to have minimum 0 and maximum 1, followed by another such normalization to ensure the high value is 1 and the low 0 after applying the sigmoid. We use $s = 10$ and redefine Eqn. 6.

$$\texttt{normalize}(\mathbf{X}) = \frac{\mathbf{X} - \min_{h,w}(\mathbf{X})}{\max_{h,w}(\mathbf{X}) - \min_{h,w}(\mathbf{X})} \tag{14}$$

$$\mathcal{A}^{\text{thresh}} = \texttt{normalize}\left(\texttt{sigmoid}\left(s \cdot (\texttt{normalize}(\mathcal{A}) - 0.5)\right)\right) \tag{15}$$

$$\texttt{size}(k) = \frac{1}{HW} \sum_{h,w} \mathcal{A}^{\text{thresh}}_{h,w,k} \tag{16}$$

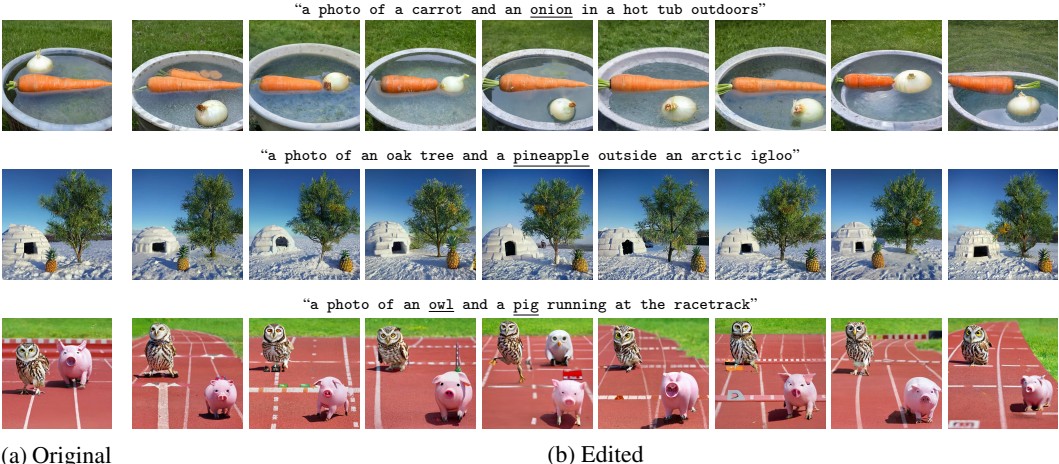

"a photo of a carrot and an onion in a hot tub outdoors"

"a photo of an oak tree and a pineapple outside an arctic igloo"

"a photo of an owl and a pig running at the racetrack"

(a) Original                             (b) Edited

Figure 11: **Moving objects.** Non-cherry-picked results for moving objects in scenes using Eqn. 9. We move `onion` down and to the right, `pineapple` to the right, and `owl` up and `pig` down, respectively. All scenes use weights $w_0 = 1.5$, $w_1 = 0.25$, and $w_2 = 2$.

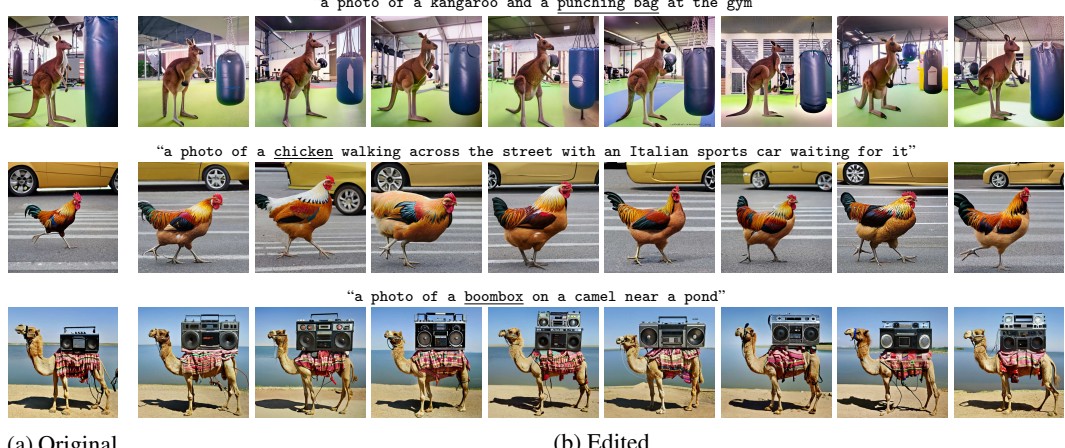

"a photo of a kangaroo and a punching bag at the gym"

"a photo of a chicken walking across the street with an Italian sports car waiting for it"

"a photo of a boombox on a camel near a pond"

(a) Original                  (b) Edited

Figure 12: **Resizing objects.** Non-cherry-picked results for resizing objects in scenes using Eqn. 9, with $\mathcal{T}$ specified to up- or down-sample attention maps. We reduce the `punching bag`'s height $0.5\times$ and enlarge `chicken` $2.5\times$ and `boombox` $2\times$. All scenes use $w_0 = 2$, $w_1 = 0.25$, and $w_2 = 3$.

## B. Using self-guidance

**Maximizing consistency.** In general, we find that sharing the same sequence of noise in the DDPM process between an image and its edited version is *not necessary* to maintain high levels of consistency, but can help if extreme precision is desired. We find that maintaining object silhouettes under transformations such as resizing and repositioning is more effective if applying a transformation $\mathcal{T}$ to the original `shape`, rather than expressing the same change through `centroid` and `size`.

**Guiding "background" words.** To keep all objects of the scene fixed but one (Fig. 3), one can either guide all other tokens in the prompt (including "a photo of" and other abstract terms) to keep their shape, or only select the other salient objects and hold those fixed. In general, since abstract words are often used for message passing and have attention patterns that are correlated with the layout of the scene, we prefer not to guide their layouts to maximize compositionality.

**Mitigating appearance-layout entanglement.** When words or concepts span multiple tokens, we can mean-pooling attention maps across these tokens before processing them, though do not find this to improve results. We also find that corrupting target shapes with Gaussian noise helps mitigate this effect, providing some evidence for this hypothesis.

**Moving objects.** We use $w_0 \in [0.5, 2]$, $w_1 \in [0.03, 0.3]$, $w_2 \in [0.5, 5]$ in Eqn. 9. Alternatively, we can express $o_k$'s new location through its `centroid`, adding a term to keep `size` fixed:

$$
\begin{aligned}
g = w_0 &\overbrace{\frac{1}{|O|-1} \sum_{o \neq o_k} \frac{1}{|\mathcal{A}|} \sum_{i=0}^{|\mathcal{A}|} \|\texttt{shape}_{i,t,\text{orig}}(o) - \texttt{shape}_{i,t}(o)\|_1}^{\text{Fix all other object shapes}} \\
+ w_1 &\overbrace{\frac{1}{|O|} \sum_{o \in O} \|\texttt{appearance}_{t,\text{orig}}(o) - \texttt{appearance}_t(o)\|_1}^{\text{Fix all object appearances}} \\
+ w_2 &\overbrace{\frac{1}{|\mathcal{A}|} \sum_{i=0}^{|\mathcal{A}|} \|\texttt{size}_{i,t,\text{orig}}(o_k) - \texttt{size}_{i,t}(o_k)\|_1}^{\text{Fix } o_k\text{'s size}} \\
+ w_3 &\underbrace{\frac{1}{|\mathcal{A}|} \sum_{i=0}^{|\mathcal{A}|} \|\texttt{target\_centroid} - \texttt{centroid}_{i,t}(o_k)\|_1}_{\text{Change } o_k\text{'s position}}
\end{aligned}
\tag{17}
$$

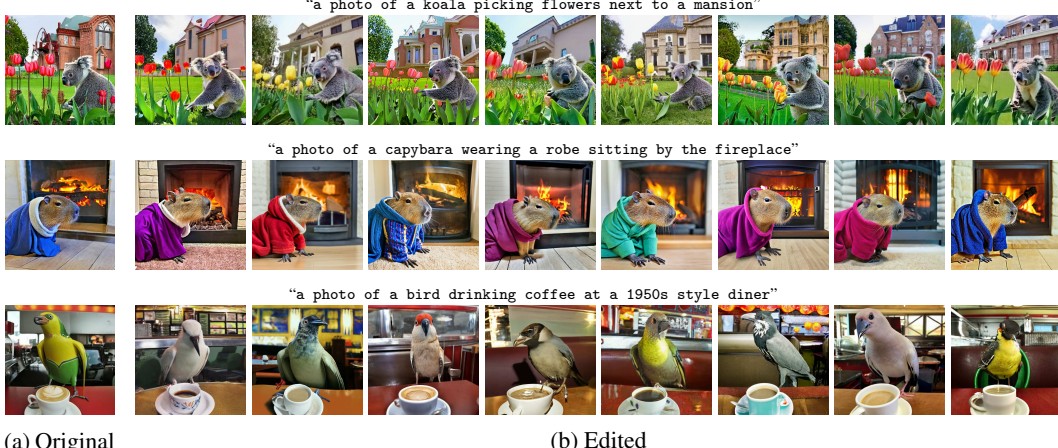

(a) Original                                 (b) Edited

Figure 13: **Creating new appearances for scenes.** Non-cherry-picked results sampling different "styles" of appearances given the same layout, using Eqn. 10. We use $w_0 = 0.7, 0.3$, and $0.3$ respectively for each result, to preserve greater structure in the background of the first picture.

Where `target_centroid` can be computed as a shfited version of the timestep-and-attention-specific `centroid`$_{\text{orig}}$ if desired, or selected to be an absolute value on the canvas (repeated across all timesteps). We generally use weights $w_0 \in [0.5, 2], w_1 \in [0.03, 0.3], w_2 \in [0.5, 2], w_3 \in [1, 3]$.

**Resizing objects.** We can follow Eqn. 9 to resize objects as well, by setting $\mathcal{T}$ to upsample or downsample the original mask. We can similarly use Eqn. 17, omitting the final term and setting the target `size` to a desired value, either computed as a function of `size`$_{\text{orig}}(o_k)$ or provided as an absolute proportion of pixels on the canvas that the object should cover. We use the same weight range for all weights except we set $w_2 \in [1, 3], w_3 = 0$ for Eqn. 17.

**Sampling new appearances.** We set $w_0 \in [0.1, 1]$ in Eqn. 10. Generally, higher values lead to extremely precise layout preservation at the expense of diversity in appearance.

**Sampling new layouts.** Just as we can find new appearances for a scene of a given layout, we can perform the opposite operation, finding new layouts for scenes where objects have a given appearance:

$$g = w_0 \overbrace{\frac{1}{|O|} \sum_{o \in O} \|\texttt{appearance}_{t,\text{orig}}(o) - \texttt{appearance}_t(o)\|_1}^{\text{Fix all appearances}} \tag{18}$$

We almost always use $w_0 \in [0.05, 0.25]$.

**Collaging objects in-place.** Eqn. 12 can be easily generalized to more than one object per image (adding another sum across all objects) or to the case where prompts vary between images (mapping from $k_j$ to the corresponding indices in the new image). We set $w_0 \in [0.5, 1], w_1 \in [0.05, 0.3]$.

**Collaging objects with a new layout.** As shown in Fig. 6f, we can also collage objects into a new layout specified by a target image $J + 1$, in addition to the $J$ images specifying object appearance:

$$g = w_0 \frac{1}{|\mathcal{A}|} \overbrace{\sum_{i=0}^{|\mathcal{A}|} \|\texttt{shape}_{i,t,J+1}(o_{k_{J+1}}) - \texttt{shape}_{i,t}(o_k)\|_1}^{\text{Copy all object shapes}}$$

$$+ w_1 \frac{1}{J} \underbrace{\sum_{j} \|\texttt{appearance}_{t,j}(o_{k_j}) - \texttt{appearance}_t(o_k)\|_1}_{\text{Copy each object's appearance}} \tag{19}$$

As in Eqn. 12, we set $w_0 \in [0.5, 1], w_1 \in [0.05, 0.3]$.

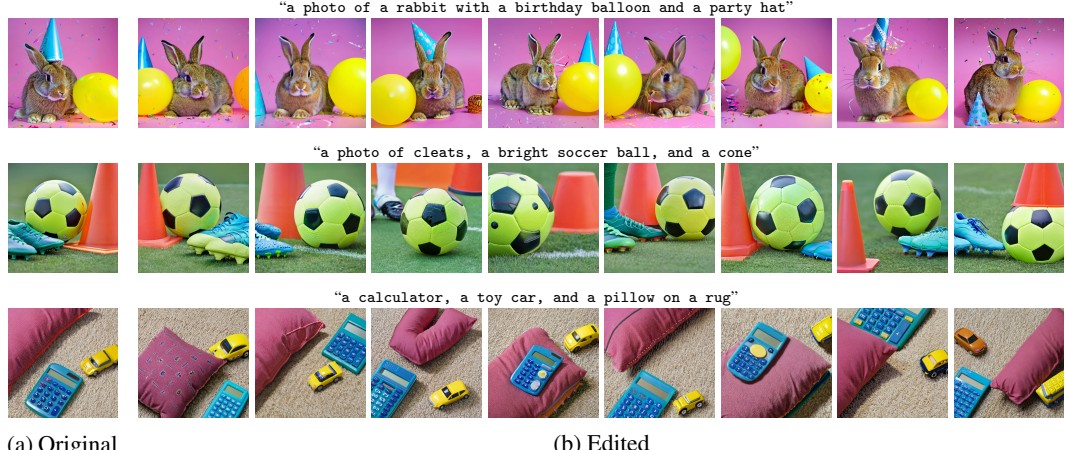

(a) Original  (b) Edited

Figure 14: **Creating new layouts for scenes.** Non-cherry-picked results sampling new layouts for the same scenes, using Eqn. 18. We use $w_0 = 0.07$, $0.07$, and $0.2$ respectively.

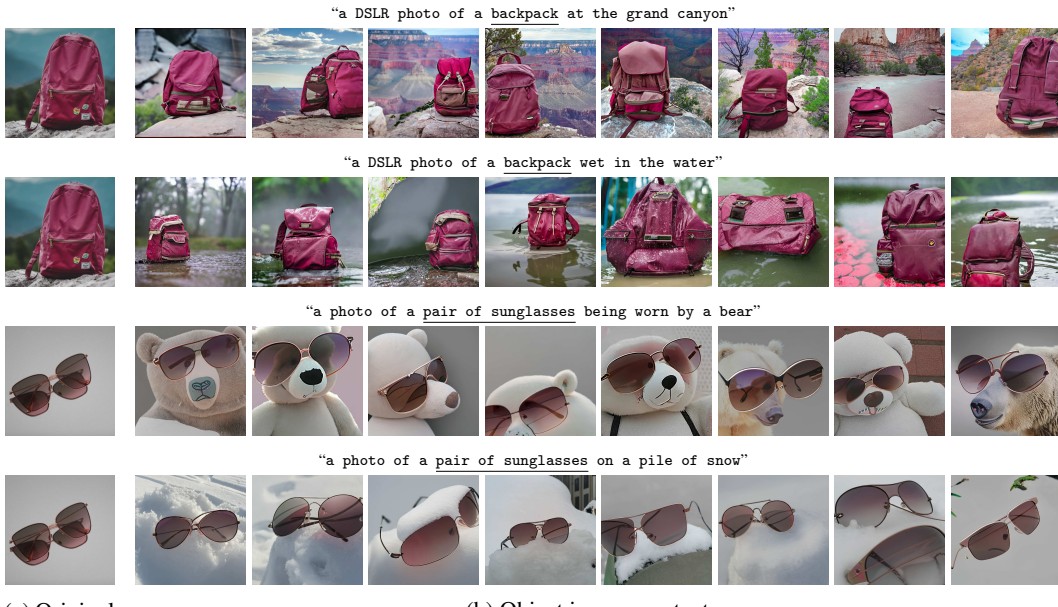

(a) Original  (b) Object in new contexts

Figure 15: **Appearance transfer from real images.** Non-cherry picked results sampling new images with a given object's appearance specified by a real images, as in Eqn. 13. We use $w_0 = 0.15$.

**Transferring object appearances to new layouts.** Nothing requires the indices (or in fact, the objects those indices refer to) to be the same in the image being generated and the original image being used as a source, as long as there is a mapping specified between the indices in the old and new images which should correspond. Call this mapping $m$. We can then take the appearance of an object $o_k$ in a source image and transfer it to an image with any new prompt as follows, as specified in Eqn. 13 (with typical weights $w_0 \in [0.01, 0.1]$):

$$g = w_0 \overbrace{\|\texttt{appearance}_{t,\text{orig}}(o_k) - \texttt{appearance}_t\left(m(o_k)\right)\|_1}^{\text{Copy object appearance}} \tag{20}$$

**Merging layout and appearance.** We use $w_0 \in [1, 2]$ and $w_1 \in [0.1, 0.3]$ in Eqn. 11.

**Editing with real images.** Importantly, our method is not limited to editing generated images to whose internals it has access by definition. We find that we can also meaningfully guide generation

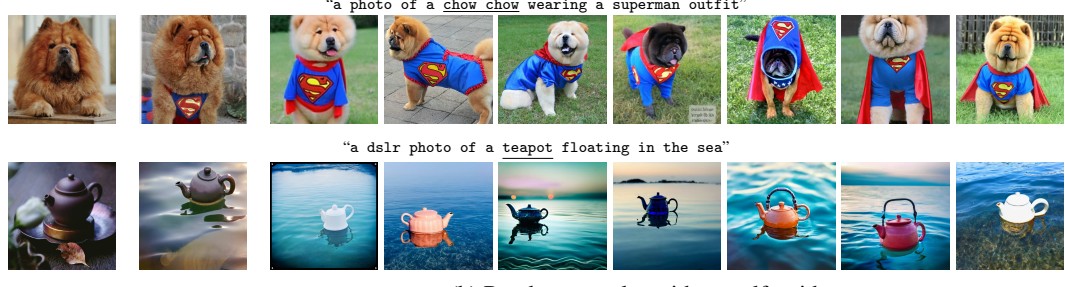

(a) Original     (b) Ours                  (b) Random samples without self-guidance

Figure 16: **Ablating appearance transfer from real images.** To verify the efficacy of our approach, we compare our results from Fig. 7 in the paper to random samples from the same prompt without apperance transfer. We can see that appearance of objects varies significantly without self-guidance.

using the attention and activations extracted from a set of forward-process denoisings of a real image (given a caption) to "approximate" the reverse process, despite any mismatch in distributions one might imagine. Concretely, we generate $T$ corrupted versions of a real image $x$, $\{\alpha_t x + \sigma_t \epsilon_t\}_1^T$, where $\epsilon_t \sim \mathcal{N}(0, 1)$. We then extract the attention $\mathcal{A}_t$ and activations $\Psi_t$ from the denoising network at each of these timesteps in parallel and concatenate them into a length-$T$ sequence. We treat this sequence identically to a sequence of $T$ internals given by subsequent sampling steps, and can thus transfer the appearance of objects from real images, output images that look like real images with moved or resized objects, and so on.

In Fig. 7, the prompts we use to transfer appearance are "A photo of a Chow Chow..." and "A DSLR photo of a teapot...". While our method works on less specific descriptions as well, it is not as reliable when object appearance is more out-of-distribution. For context, we show unguided samples under the prompts from Fig. 7 in Fig. 16, which still deviate significantly from the desired appearance, showing the efficacy of our approach. A weakness of our simple approach is that it has no constraints on the shape of the generated objects, which we leave to future work.

**Weight selection heuristics.** We find weights that work well to remain more or less consistent across different images given an edit, but ideal weights do vary somewhat (within predictable ranges) between different combinations of terms. Our heuristics for weight selection per term are: the more weights there are, the higher per-term weights can be without causing artifacts (and indeed, need to be, to provide ample contribution to the final result); `appearance` terms should have weights 1 or 2 orders of magnitude lower than layout terms; layout summary statistics (`centroid` and `size`) should have slightly lower weights than terms on the per-pixel `shape`; total weight of terms should not add up to more than $\sim 5$ to avoid artifacts.

## C. Additional results

We show further non-cherry-picked results for the edits we show in the main paper. Our general protocol consists of selecting an interesting prompt manually, verifying that our model creates compelling samples aligning with this prompt without self-guidance, beginning with the typical weights we use for an edit, and trying around 3-5 other weight configurations to find the one that works best for the prompt – in most cases, this is the starting set of weights. Then, we use the first 8 images we generate, without further filtering. We generate all results with different seeds to showcase the strength of guidance even without shared DDPM noise. We show more results for moving (Fig. 11) and resizing (Fig. 12) objects, sampling new appearances for given layouts (Fig. 13) as well as new layouts for a given set of objects (Fig. 14), and transferring the appearance of real objects into new contexts (Fig. 15). We also include an ablation on hyperparameter values (Fig. 17) as well as preliminary results of an implementation of self-guidance on an open-source diffusion model in Fig. 18.

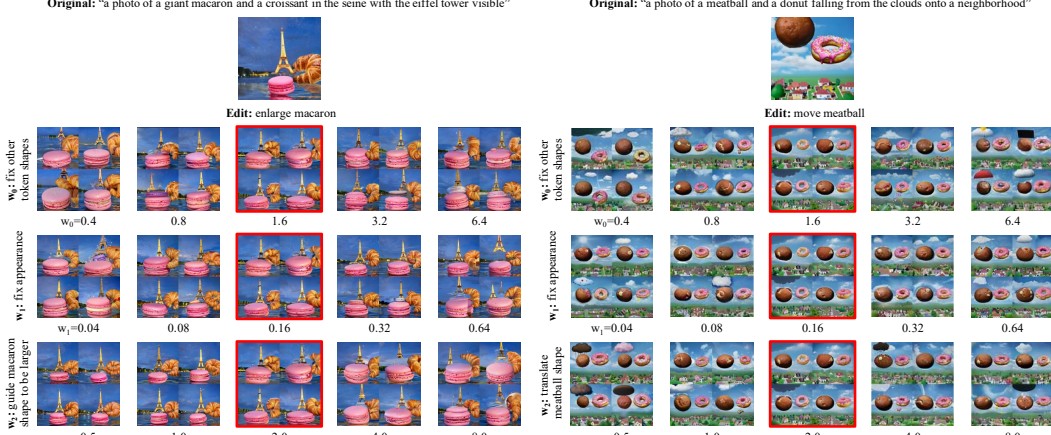

Figure 17: **Hyperparameter sweeps.** We show results for two edits (moving and resizing objects) using Eqn. 4 in the Supp. Mat., for different values of weights on the three edit terms, holding the other terms to the value in the middle column. Please zoom in to view in more detail. Reasonable values in the middle columns (within the expected range) lead to overall successful image manipulation. Very large hyperparameter values cause visual artifacts to appear (by moving sampling off-manifold) while still tending to perform the edit successfully, while extremely small values often fail to conduct the edit, inducing artifacts resulting from a "half-executed" manipulation.

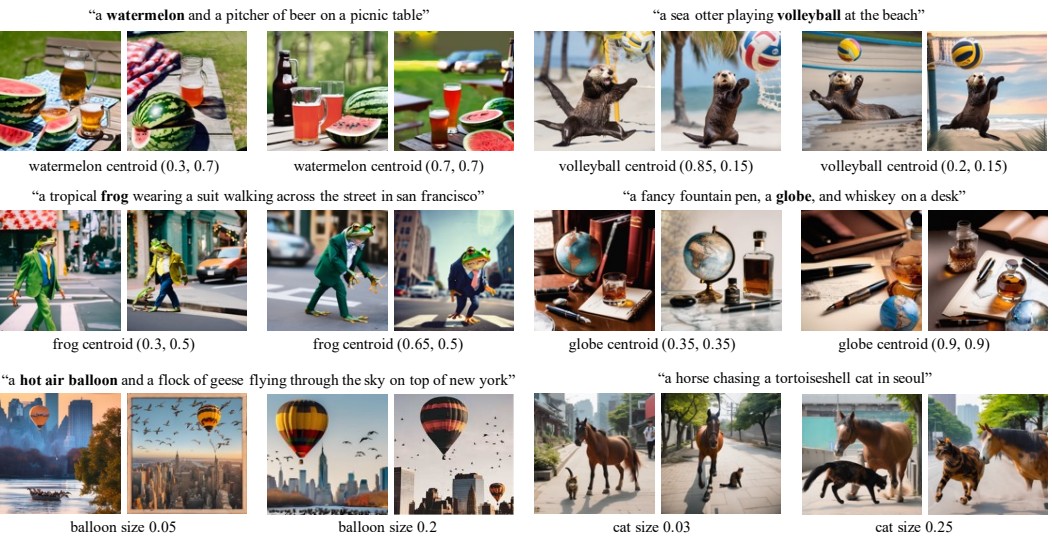

Figure 18: **Self-guidance on Stable Diffusion XL.** To highlight the generality of our approach, we demonstrate preliminary results for controllable generation on a popular latent-space text-to-image diffusion model, using 100 DDPM steps (applying self-guidance from step 10 to step 90). We get best results only guiding attention in the second decoder block of the denoiser model.

