# OpenReview forum: "Diffusion Self-Guidance for Controllable Image Generation"
_NeurIPS.cc/2023/Conference — NeurIPS 2023 poster_

### Official Review · Reviewer_qibF · 2023-07-05

**Soundness:** 3 good
**Presentation:** 4 excellent
**Contribution:** 3 good
**Rating:** 7
**Confidence:** 4

**Summary:**

The paper introduces a new method for detailed and versatile image editing and controlled image synthesis with large-scale text-to-image diffusion models. In particular, the authors propose self-guidance: Self-guidance uses the internal representations of the denoiser neural network of diffusion models for guidance. The work finds that the attention and feature maps of the denoiser network encode properties like objects' size, shape, position and appearance in an interpretable way. This is, some heuristics on the internal attention/feature maps can be defined that represent these properties well and then we can do guidance, leveraging these internal representations (note that this works only for objects and words that were mentioned in the text prompts driving generation). Hence, given an image (either generated by the diffusion model itself or a real one, reconstructed by the diffusion model), the novel self-guidance technique enables editing operations such as freely modifying objects' size, shape, position and appearance, without introducing noticeable artifacts. Importantly, this does not require any additional training or use of auxiliary models, and no additional paired data for training, which makes this a very powerful technique. The authors convincingly validate their proposed editing method mostly qualitatively by showing lots of interesting editing examples.

**Strengths:**

Generally, the main strength of the paper is the introduction of the novel self-guidance technique, which it then uses for detailed and advanced image editing and controllable image synthesis. To the best of my knowledge no previous image editing methods with text-to-image diffusion models reach this level of fine control with respect to objects' shape, size, position, appearance, layout, and composition, while introducing almost no artifacts.

**Clarity:** The paper is very well written and easy to follow and understand. It is very clear.

**Originality:** The main idea of self-guidance leveraging the model's internal representations is novel and original. That said, it is closely related to, and probably inspired by, existing technique such as Prompt-to-prompt editing (Hertz et al. "Prompt-to-Prompt Image Editing with Cross Attention Control") and paint-with-words (Balaji et al. "eDiff-I: Text-to-Image Diffusion Models with an Ensemble of Expert Denoisers").

**Significance:** Controllable image generation and image editing with large-scale text-to-image diffusion is an important direction with many relevant applications. The paper proposes a very powerful and versatile method with little requirements, which makes this a significant contribution.

**Quality:** The overall quality of the paper is high. It is clear, well written, all claims are supported, limitations and societal impacts are appropriately discussed, background is provided, and related work is thoroughly discussed.

**Weaknesses:**

While the paper has few major issues, there are some concerns:
- Editing and control is limited with respect to objects (and words more generally) that appear in the text prompt. This also means that given real images require an appropriate caption to be editable. Note that this limitation is acknowledged by the authors.
- The method comes with a ton of hyperparameters: The guidance weights for the different edits need to be chosen manually and it seems like some trial and error can be necessary. Also, more detailed ablations on this would be helpful, like showing different guidance weights for same image/seed/prompt/edit, etc.
- When performing a simple edit like moving a single object, we actually require a lot more guidance terms for keeping all the other objects in place and preserving their appearance. This means that even edits that may appear simple are actually quite complicated and require guidance weights for all different terms.
- The way the different edits and controls are defined is largely heuristic (how size, shape, appearance, etc. is calculated from the attention/feature maps). It seems to work well, but in theory it would be great if there was a more principled approach to this.
- The way the cross-attention maps are leveraged in self-guidance is related to the "paint-with-words" technique introduced in eDiff-I (Balaji et al. "eDiff-I: Text-to-Image Diffusion Models with an Ensemble of Expert Denoisers"). I believe this should be acknowledged and cited.
- The approach is validated only in pixel-space text-to-image diffusion models. Can the same method also work in latent diffusion models? The most popular, and publicly available, large-scale text-to-image model is Stable Diffusion. It would be great to see the method also applied there, if possible.

In conclusion, there are concerns, weaknesses and routes for further improving the paper. However, the strengths outweigh these weaknesses. Consequently, I am leaning towards suggesting acceptance of the paper. I am looking forward to the rebuttal by the authors.

**Questions:**

I also have a variety of questions that did not significantly impact the rating of the paper:
- I believe the paper intrinsically relies on a typical U-Net architecture for the denoiser network that gives rise to the useful attention and feature maps. There are various recent works that use pure vision transformer denoiser networks or advanced architectures like recurrent interface networks. Can the approach be extended to these architectures, too? I would be interested in the authors' thoughts on this.
- In a similar spirit, do the authors believe that similar techniques can also be used in diffusion models that model video, 3D, audio, graph/molecule data, etc.?
- While centroid, size and shape leverage all cross attention maps, the appearance term uses only the activations of the penultimate layer in the decoder as well as the final cross-attention operation. Why is that? Can the authors provide more intuitions for the qualitative differences in how the attention/activations are used in the different operations?
- The paper entirely leverages DDPM-style sampling. Does the method also work with deterministic probability flow/DDIM sampling? Moreover, when using given real images, could there be an alternative way of editing when probability flow ODE/DDIM is used to deterministically encode a given image and then re-generate again while introducing the desired edits? I would be curious about the authors' perspective on this.

Very Minor nitpick:
- Please properly introduce the data $x$ as well as the noise schedule $\alpha_t$, $\sigma_t$ around equation 1.
- I would suggest to properly introduce and define the indices in Line 155 and below, this is, what exactly is $i$, etc. to avoid any potential confusion.

**Limitations:**

Limitations and potential negative societal impact have been appropriately discussed and are well addressed. There are no concerns in that regard.

---

> ### Author Rebuttal · Authors · 2023-08-10
>
> Thanks very much for the in-depth reading of our paper and for the helpful comments and suggestions.
>
> **Limitation to entities mentioned in prompt:** Yes, this is indeed a primary limitation of self-guidance as presented. However, other forms of self-guidance could help in situations where a text description is not available. For example, to move objects, one can select a point on the canvas to extract the feature vector at that location and guide that feature to appear at another location instead; one could also extract a soft segmentation map by finding feature vectors (which have been shown in concurrent work to be meaningful, e.g. [1,2]) that are epsilon-close to the “clicked” feature, rather than by going through the cross-attention map, and using those as self-guidance targets (e.g. to resize objects or change their appearance). We leave fuller exploration of these ideas to future work, though preliminary results indicate the direction is promising.
>
> **Sensitivity to hyperparameters:** We agree that more discussion on sensitivity to hyper-parameters is valuable. As also discussed in responses to reviewers EBx5 and jH2E, we are releasing an open-source implementation of the model so that the community can explore these aspects, but we also find that self-guidance is not highly sensitive to choices of weights, etc. and that once good settings for hyperparameters are found, they can remain more-or-less fixed across images. We also plan on adding a figure demonstrating the effect of varying these axes to the final manuscript, and included a figure in the global response to the reviews visualizing this.
>
> **Self-guidance terms:** We agree that further exploration of self-guidance and a more theoretically grounded approach is an exciting future direction. Though the terms we propose are indeed “heuristic”, they proved effective for the desired edits on the models we explored. The mechanistic role of attention interactions in large models remains an open research question and we see self-guidance as another tool that can expose unexpected behaviors in these models, hopefully leading to a better understanding of their internals. For example, in Figure 9 (c,d), we find hints that – somewhat counterintuitively – spatial attention patterns themselves may somehow be used to encode desired appearance information of objects.
>
> **Pixel-space vs latent-space models:** We do find that self-guidance generalizes to Stable Diffusion and are releasing an open-source implementation of such, please see global comment for more detail.
>
> **Self-guidance across architectures and modalities:** The core idea behind self-guidance is that powerful generative models learn interesting internal representations, and we can use these representations to guide sampling. This does not rely on attention, but does rely on a method for identifying useful features from internals and a way to guide sampling. We are hopeful that similar techniques can be useful for other modalities.
>
> **Appearance term:** This is a fair point. Empirically, we found that guiding other features in the UNet did not control appearance (at least as perceived by humans) as much as the features toward the very end of the model, nearest to the output head. Attention maps at that stage are also higher-resolution with sharper edges. This motivated the formulation of a single appearance term rather than a bundle throughout the depth of the network. It is possible that we could effectively control other attributes by only guiding one attention layer as well, and this may allow a modest speedup at inference time.
>
> **Other sampling methods:** We found self-guidance to work somewhat worse with DDIM. We hypothesize this is due to the higher stochasticity of DDPM being less sensitive to going off-manifold or perhaps due to the inaccuracy intrinsic to the reverse diffusion sampler [3]. That being said, in the Stable Diffusion implementation, we are able to use other schedulers such as the Euler discrete sampler [4] with success. Recent work has shown the ability to reconstruct images by inverting multiple layers of DDPM noise [5], and this may indeed be a promising avenue to further improve quality of real image editing.
>
> **Minor nitpicks:** Thanks for the attentive reading of the manuscript and we will incorporate these changes into its final edition, including citing eDiff-I and defining indices and variables in equations.
>
> * [1] Diffusion Hyperfeatures: Searching Through Time and Space for Semantic Correspondence, Luo et al., 2023
> * [2] Emergent Correspondence from Image Diffusion, Tang et al., 2023
> * [3] Reduce, Reuse, Recycle: Compositional Generation with Energy-Based Diffusion Models and MCMC, ICML 2023
> * [4] Elucidating the Design Space of Diffusion-Based Generative Models, Karras et al., NeurIPS 2022
> * [5] An Edit Friendly DDPM Noise Space: Inversion and Manipulations, Huberman-Spiegelglas et al., 2023

---

> > ### Comment · Reviewer_qibF · 2023-08-18
> > **Thank you for rebuttal**
> >
> > I would like to thank the authors for their rebuttal and for replying in detail to all my questions. I do not have any further questions. It's great to see that the method also seems to work in Stable Diffusion. The additional results in the pdf are also helpful and make me a bit less concerned about hyperparameters (although that's still a weakness). Overall, I believe this paper should be accepted. I raised my score by one point.

---

### Official Review · Reviewer_hErC · 2023-07-06

**Soundness:** 2 fair
**Presentation:** 3 good
**Contribution:** 2 fair
**Rating:** 5
**Confidence:** 4

**Summary:**

Large-scale text-to-image generative methods have demonstrated impressive performance given a detailed text prompt. However, many aspects of an image are hard or impossible to specify with texts. The authors proposed self-guidance to guide the generation of diffusion models. Experimental results demonstrated that object size, location, and appearance of objects can be controlled from normalized attention matrices and feature activations.

**Strengths:**

1. The motivation and background is explained clearly. Arguments and claims are supported by references or experimental results.
2. The proposed approach is simple and effective. The presented empirical results on multiple tasks demonstrated the effectiveness of the approach. The authors claimed that all results are non-cherry-picked, which is quite impressive.
3. Adding geometrical/compositional control to the image generative models is an interesting and important research topic. The authors proposed a simple and smart method to control existing generative models with diffusion guidance.
4. Multiple tasks were considered, including adjusting individual properties, compositional generation, and real image editing. This proposed approach showed potentials to a range of real-world applications.

**Weaknesses:**

1. From the abstract, the authors aimed to add more control over the image generation beyond text prompts. However, the proposed approach is not a good solution in this regard – it is strongly based on the tokens in the text prompt. There are two limitations:
   * Can we control fine-grained objects/parts not naturally present in the text prompt? For example, control the dog mouth when generating “a photo of a dog wearing a baseball cap”? All current results are on the object-level with no finer control.
   * The proposed approach is limited to objects/parts easily describable by texts. This is a disadvantage compared to the keypoint inputs from DragGAN. For instance, annotating a keypoint would be more straightforward than inputting “left-front leg of the dog”, for both the user and the model.
2. Based on the framework, designing more control seems difficult, e.g., 3D viewpoint of objects. Compared to parallel works, ControlNet [2] finetuned the LDM model with 3D representations and DragGAN [1] used finer keypoint inputs to accomplish this.
3. The proposed method is built on existing large-scale image diffusion models. It heavily relies on a good representation learned by these trained models. The aforementioned limitations are mostly due to the latent representations having limited feature disentanglement or 3D awareness. The proposed approach doesn’t seem to have any easy fixes to address these limitations.

References:
1. X. Pan et al. Drag Your GAN: Interactive Point-based Manipulation on the Generative Image Manifold.
2. L. Zhang et al. Adding conditional control to text-to-image diffusion models.

**Questions:**

1. To see how entangled the representations are, can the authors generate a video or a dense sampling of intermediate images to interpolate the “control space”? For instance, changing the layout of the objects in Figure 5 over 16 frames, or changing the styles in Figure 4 over 16 frames? In this way we could see: (i) if a continuous control corresponds to smooth and continuous changes in the image space, and (ii) how the other objects are influenced by the controlled changes made to the target object.

**Limitations:**

1. The input space of the proposed control is still limited by texts: see weakness 1.
2. The proposed approach is haunted by entangled representation: see weakness 2 and 3.

---

> ### Author Rebuttal · Authors · 2023-08-10
>
> Thank you for your time and feedback on our work.
>
> We respectfully disagree that self-guidance is not a “good solution” to “add more control over image generation beyond text prompts”. The edits that we perform, such as moving or resizing objects, are very challenging (if not impossible) to affect through text, especially if consistency with a source image is desired. Indeed, no previous method has shown this functionality. We acknowledge in the manuscript that self-guidance as we present it requires entities to be mentioned in the text prompt, but please also see discussion under “Limitation to entities mentioned in prompt” in the response to reviewer qibF.
>
> Self-guidance enables a wide variety of image manipulations that – though they rely on text-image attention – *are not themselves feasible just through prompting with text*. Concurrent work such as DragGAN has been implemented in diffusion models already, in fact, building on self-guidance [1], highlighting the strength and versatility of our approach. 3D viewpoint control is an interesting future direction, but we restrict our attention to 2D for this work. The fact that self-guidance relies on the strong representations learned by text-to-image models is a feature, not a bug – we show that many complex image manipulations require *no additional fine-tuning or supervision or auxiliary models* (which ControlNet does). We wish to highlight that many of the manipulations enabled by self-guidance have not been shown at all in the literature for diffusion models previously.
>
> [1] DragonDiffusion: Enabling Drag-style Manipulation on Diffusion Models, Mou et al., 2023

---

> ### Comment · Reviewer_hErC · 2023-08-14
> **Final Rating**
>
> The authors proposed a simple but effective approach for controllable image generation/editing. However, I believe there are two major limitations of this work:
>
> 1. Inconsistent editing (EBX5) or feature entanglement (hErC): a simple observation can be made from the qualitative examples that as we apply control to one object, other objects/background may change. This issue can be more severe as multiple controls are involved.
> 2. Control limited to existing entities in the prompt (qibF) and cannot achieve fine-grained control (e.g., parts of the object) (hErC).
>
> Regarding the first issue, the authors claimed that a superior approach can be adopted by sharing DDPM noise between the two images (as described in Supp. Mat L27-29), and the second limitation can be solved by annotating keypoints on the image and then applying control.
>
> It should be noted that both approaches are not verified or carefully tested. Sharing DDPM noise may have been tested on a few examples but a thorough test is necessary if this is claimed to address the consistency limitation. Using keypoints as a control is another highly risky idea as the proposed approach may not naturally work or requires a lot of effort to adapt.
>
> I believe the two issues significantly limit how widely and effectively the proposed approach can be applied. A thorough analysis of these limitations is necessary in the main text of the paper as understanding the strengths and weaknesses are both crucial for a sound paper.
>
> Overall I think this work is limited in a few aspects with direct or indirect comparison with parallel methods. All initial reviews lean towards acceptance but I believe the current scores is a bit overrated given these concerns above.

---

> > ### Author Response · Authors · 2023-08-15
> > **Response to reviewer**
> >
> > Thank you for the continued engagement to improve our work. We openly discuss the editing limitations you mention (Sec 5, Fig 9), and agree that further work is needed to improve our method in the editing context. In many controllable generation settings (e.g. layout, centroid, and size conditioning, zero-shot DreamBooth), these limitations do not apply, and regardless, we are not aware of any prior work that has achieved the same breadth of capabilities without supervision (e.g. moving objects, resizing them, copying appearance from a real image into a generated one). If you are aware of any other existing methods with the same functionality enabled by self-guidance that do not require training, we are happy to include them in our paper. We attempted to provide comparisons wherever possible to other work such as DreamBooth (though it requires fine-tuning) and PromptToPrompt.
> >
> > We agree that the methods we presented for sharing noise to reduce changes in not-controlled parts of the image and our proposal to use keypoints for control are not thoroughly tested, but they are also not core components of our method (all figures in the Supp. Mat., Fig. 4, Fig. 5, do not use shared noise). The limitation of only being able to control what is described with text can be addressed by expanding the text prompt to describe more of the image. This form of control isn’t much more challenging than annotating objects with keypoints (as DragonDiffusion does, citing our work).
> >
> > As we mentioned in the response to EBx5, “we find that as long as the specified constraints do not contradict each other, the effectiveness of self-guidance does not decay as more terms are added.” Could the reviewer please point to evidence that “the issue can be more severe as multiple controls are involved” beyond this?
> >
> > Thanks very much again for your thoughtful response and reading the paper in detail.

---

> > > ### Comment · Reviewer_hErC · 2023-08-22
> > > **Re: Response to reviewer**
> > >
> > > Thanks for the response.
> > >
> > > Regarding “the issue can be more severe as multiple controls are involved”: As noted in my previous comment, this refers to the limitation of "inconsistent editing (EBX5) or feature entanglement (hErC)", such as undesired changing of background or uncontrolled objects due to other controls, which is a different topic than "the effectiveness of self-guidance (on controlled properties)". I don't find any detailed analysis of this limitation, quantitatively or qualitatively.
> > >
> > > I acknowledge the originality and effectiveness of this work but have concerns about the analysis of limitations and its future extensions. Hence I cannot grant a higher rating in its current form.

---

### Official Review · Reviewer_jH2E · 2023-07-07

**Soundness:** 4 excellent
**Presentation:** 3 good
**Contribution:** 3 good
**Rating:** 6
**Confidence:** 5

**Summary:**

This paper introduces diffusion self-guidance, an inference-time technique for controllable image generation using pre-trained text-to-image diffusion models. The key finding is that internal representations of the denoiser network carry meaningful information about the scene, and one can build custom energy functions around these representations to align image generation with user-defined scene properties, including object location, size, shape and appearance. Further, an appealing property of self-guidance is that the energy functions can be flexibly composed to support the simultaneous manipulation of multiple image attributes. Extensive experiments are conducted to demonstrate the effectiveness of self-guidance across a broad spectrum of image manipulation tasks.

**Strengths:**

- The proposed method enables controllable image generation / manipulation using existing text-to-image diffusion model checkpoints without costly fine-tuning. As a training-free method, it achieves controllability by steering the generation towards matching the desired internal representations at inference time. This guidance-based design is conceptually simple, easy to implement, computationally efficient and highly flexible.

- The method represents object properties using aggregated statistics of denoiser representations. It thus reveals the internal working of the generative process. To this end, the impact of the paper goes beyond controllable image generation / manipulation.

- Self-guidance enables several edits that are not possible with concurrent methods. These include change in object location, size and modification of scene layout. In the meantime, self-guidance allows the composition of multiple edits in a single generation pass. This is another unique property that is rarely seen in previous and concurrent works.

- Finally, self-guidance offers compelling editing capabilities and supports a wide range of image manipulation needs. Among them are the mixing of objects and layout from multiple source images, transfer of attributes defined by non-object words, and the editing of real images.

**Weaknesses:**

I do not have major concerns about the proposed method. Some minor concerns are as follows.

- The main text presents self-guidance in its generic form and provides the ingredients for defining a specific energy function. However, all concrete examples are left to the supplement. I understand that space is limited and the authors want to make room for the results (which are indeed compelling). However, it is awkward to alternate between the main text and the supplement in order to decipher how those edits are actually achieved. One possibility is to organize all instantiations of self-guidance in a big table and put them in the main text for the ease of understanding.

- I did not find ablation experiments on hyper-parameters such as the number of inference steps, guidance strength and the range of steps to apply guidance. In practice, the editing quality is often quite sensitive to these hyper-parameters. While their choices are briefly discussed in the supplement, some visual examples will help visualize their effects.

- The experiments are conducted using Imagen, whose design is substantially different from the publicly available Stable Diffusion (cascaded vs. latent diffusion). It is a well-known fact that an editing method may not work equally well on different diffusion models. Given that most players in the space of controllable image generation rely on Stable Diffusion, I would encourage the authors to showcase a few editing examples using Stable Diffusion in order to calibrate the performance of self-guidance.

**Questions:**

Please find my comments in the section above.

**Limitations:**

This paper includes a discussion on the limitations and potential societal impact of the proposed method.

---

> ### Author Rebuttal · Authors · 2023-08-10
>
> Thank you for taking the time to read and review our paper!
>
> **Paper layout:** Thanks for the feedback on this. We entirely agree and have restructured the text so that the energy functions are in the main paper rather than the supplementary material.
>
> **Sensitivity to hyper-parameters:** We always apply self-guidance on all inference steps, and find that our method works on a wide range of steps – we experimented with 128, 256, 512, and 1024. Coarsely, we find that applying self-guidance only on early steps allows for control of layout but not appearance, and only on later steps allows for control of some appearance but not layout. That being said, we agree with the review that visual demonstrations of these effects are valuable to provide a better understanding of the method’s strengths and weaknesses, and we have included this ablation figure in the global comment (and will also incorporate it into the final manuscript). Thank you for the suggestion. Please also see response to reviewer EBx5 for more discussion.
>
> **Open-source Stable Diffusion implementation:** We agree that showcasing the method’s ability to generalize beyond one specific text-to-image diffusion model is valuable and are releasing an implementation on Stable Diffusion XL. Please see the PDF in the global comment for examples of self-guidance applied to Stable Diffusion XL.

---

> > ### Comment · Reviewer_jH2E · 2023-08-18
> >
> > Thanks for the rebuttal. The clarification on hyper-parameter selection is helpful. The results on Stable Diffusion also look promising, and I am looking forward to the open-source implementation.

---

### Official Review · Reviewer_EBx5 · 2023-07-12

**Soundness:** 3 good
**Presentation:** 3 good
**Contribution:** 3 good
**Rating:** 6
**Confidence:** 4

**Summary:**

This paper proposes a method for image editing using pretrained text-to-image diffusion models. The method guides the sampling process with energy functions that are added similarly to classifier guidance. These energy functions are computed as the difference between some object's property and the target state we want to change it to. A property is either an object's location, size, shape, or appearance. These properties are represented by the cross-attention maps between the object's word token and the image, which has previously been shown to produce meaningful masks. Multiple energy functions can be composed to achieve composite edits. Experiments qualitatively demonstrate the appeal of the method.

**Strengths:**

The method is practical since it does not require retraining while still achieving impressive results (as demonstrated qualitatively with some samples).

The paper proposes a simple method with a clear and concise presentation.

The addressed problem (image editing using pretrained diffusion models) is of interest to both academics and practitioners.

**Weaknesses:**

Quantitative evaluations of the method (for example with user studies that rank/compare different editing results) would make this work more insightful. Currently, it is unclear whether the shown examples are representative of what to expect when using this method and where the limitations lie. A potential limitation that could be exposed through quantitative studies is the number of energy functions that can be composed at once. Editing real images is possible as shown in Figure 7, but it isn't clear how well it works when compared to generated images. Furthermore, from some examples (e.g., Figure 8 a-b) it looks like the non-edited parts of the image also change and do not remain consistent with the original image, as is usually desirable in image editing settings.

There are missing details that could help in reproducing and better understanding the results. See the questions part.

**Questions:**

What pretrained diffusion model is used in the experiments? Does this method perform differently for different diffusion models (Stable Diffusion vs. eDiff-I vs. Imagen vs. etc.)? I'm asking this because different diffusion models incorporate text conditioning differently (both architecturally and during training).

How sensitive is the method w.r.t. hyperparameters like the attention layer, time step, and guidance strength? Is it necessary to tune the hyperparameters for every image individually? Maybe even for every energy function individually?

**Limitations:**

Many limitations regarding the method and potential negative societal impacts are discussed. Some that might be reasonable to also mention are listed above.

---

> ### Author Rebuttal · Authors · 2023-08-10
>
> Thanks for your thorough review and thoughtful comments.
>
> **Open-source Stable Diffusion implementation:** To facilitate an even better understanding of the limitations of self-guidance, as well as address questions about whether shown examples are representative of the method’s abilities and whether our method generalizes beyond Imagen, we are releasing an implementation of self-guidance on Stable Diffusion XL. We have also attached examples of controlled generation using self-guidance on SD-XL in Fig. 2 of the PDF in the global comment.
>
> **Limitations of self-guidance:** We agree that strong numerical evaluation of controllable image generation approaches remains an open problem. We qualitatively show that our approach in quality is comparable to previous work where there is overlap in functionality (e.g. Figs. 3, 6) though it requires no fine-tuning. Additionally, our method enables many manipulations that are not possible with previous work – and we include a representative demonstration of the main shortcomings of our method in  Fig. 9. We also show that many energy functions can be composed (Fig. 4) with terms from multiple source images on appearance and layout – we find that as long as the specified constraints do not contradict each other, the effectiveness of self-guidance does not decay as more terms are added.
>
> We also show additional non-cherry-picked examples of image manipulation in the Supplementary Material, highlighting the diversity of outputs the method is capable of producing.
>
> As for editing real images, we do find a slight decrease in faithfulness to the original, which seems to stem from the limitations of the simple appearance term in our reconstruction methodology (an average-pooled per-token vector).
>
> We thank the reviewer also for pointing out slight inconsistencies in the background of the edit in Figure 8 (a,b). In this case, a higher-quality result can be obtained by sharing DDPM noise between the two images (as described in Supp. Mat L27-29), a technique we did not apply to results in that figure .
>
> **Sensitivity to hyper-parameters:** We use Imagen in all experiments shown in the paper, but our method only makes the assumption that a cross-attention interaction between text and image tokens exists in the architecture (which is the case for all current SOTA text-to-image systems). We guide all attention layers and timesteps across all edits, without per-edit tuning. Each energy function has its own “default value” weight that works well for it, and this value is not dependent on the prompt or image at hand. When composing many energy functions, one may wish to slightly increase or decrease the strength of one term when compared to others, but we find that the range of per-term weights across images is small and that any value within the range works reliably. We find hyperparameter ranges by running an ad-hoc binary search once for each self-guidance term (e.g. centroid, size, …): too-large values cause visible artifacts in image quality while small values fail to cause the edit to take place. We include details on these hyperparameters in Supp. Mat. (e.g. line 53, 58) as well as a visual demonstration of their effects in the one-page PDF attached to the global response to all reviewers.

---

### Author Rebuttal · Authors · 2023-08-10

We thank all reviewers for their time and detailed reading of our paper. Reviewers found our approach to controllable image generation “conceptually simple, easy to implement, computationally efficient and highly flexible” with “clear and concise presentation” in an “interesting and important research topic” with “many relevant applications”. Reviewers appreciated that, by guiding properties of the attention and features of large generative models, our method “does not require retraining while still achieving impressive results” and provides a level of control that “no previous image editing methods with text-to-image diffusion models reach”.

A shared concern was the sensitivity of our method to hyperparameters, namely the guidance scale that effectively weights each self-guidance term differently. We include a figure (Fig. 1) in the PDF attached to this response to demonstrate the robustness of self-guidance to different choices of scale hyperparameters. Note that self-guidance performs reasonably for a wide range of values, but extremely small or large scales induce artifacts or fail to execute the edit. Some reviewers were curious whether our approach generalizes to latent-space text-image diffusion models such as Stable Diffusion. We plan on releasing an open-source implementation of self-guidance on Stable Diffusion XL. We are still working on this implementation, but have included preliminary results in Fig. 2 of the attached PDF, validating that our method indeed generalizes to other diffusion architectures. We are also sharing an anonymized Colab notebook containing the current state of the code used to generate this figure, and are working hard to implement the remaining pieces.

---

### Decision · Program_Chairs · 2023-09-21

**Decision:**

Accept (poster)

**Comment:**

This paper proposes a method for editing images using text-to-image generative diffusion models. The proposed method guides the sampling process and can guide the object's location, size, shape or appearance. The reviews highlighted the proposed methods simiplicity, novelty and practicality and strong performances. While some points about hyperparameters and results on stable diffusion were raised by the reviewers initially, they were addressed to satisfaction of reviewers in the rebuttal.